# Urine Progesterone Level as a Diagnostics Tool to Evaluate the Need for Luteal Phase Rescue in Hormone Replacement Therapy Frozen Embryo Transfer Cycles

**DOI:** 10.3390/ijms262110795

**Published:** 2025-11-06

**Authors:** Linette Yde Hansen, Takeshi Fujisawa, Betina Boel Povlsen, Rita Jakubcionyte Laursen, Mette Brix Jensen, Peter Humaidan, Birgit Alsbjerg

**Affiliations:** 1University Clinic for Fertility, Skive Regional Hospital, 7800 Skive, Denmark; 2BHF Centre for Cardiovascular Science, The Queen’s Medical Research Institute, University of Edinburgh, Edinburgh EH16 4TJ, Scotland, UK; 3Department of Clinical Medicine, Aarhus University, 8000 Aarhus, Denmark

**Keywords:** urine progesterone, hormone replacement therapy, frozen embryo transfer, progesterone administration

## Abstract

Additional progesterone administration during the luteal phase enhances reproductive outcomes in Hormone Replacement Therapy Frozen Embryo Transfer (HRT-FET) cycles in patients with low serum progesterone (P4). In this study we wanted to explore the use of urine P4 as a diagnostic tool during the luteal phase. This prospective observational cohort included a total of 464 HRT-FET cycles. The protocol entailed oral oestradiol (6 mg/24 h), followed by vaginal micronised progesterone (400 mg/12 h). On the day of blastocyst transfer, urine and serum samples were collected. Urine samples were analysed using an ARCHITECT automated immunoassay. A significant difference was found in median urine P4 between patients with serum P4 higher or lower than 11 ng/mL: 6400 ng/mL IQR [2528; 11,930] vs. 3408 ng/mL IQR [592; 6688], *p* < 0.001. The optimal cut-off to achieve live birth was a urine P4 ≥ 4000 ng/mL. The live birth rate was significantly higher in patients with urine P4 ≥ 4000 ng/mL, 48% (107/222) vs. 35% (45/130), respectively (*p* = 0.013). The odds ratio for live birth was 1.8 in patients with urine P4 ≥ 4000 ng/mL, 95% CI [1.067; 3.018], *p* = 0.028. The findings of the present study suggest that urine progesterone could be a valuable diagnostic tool to evaluate the need for additional progesterone in HRT-FET cycles.

## 1. Introduction

Current evidence shows that low serum progesterone (P4) levels adversely affect reproductive outcomes in both true Natural Frozen Embryo Transfer (NC-FET) cycles and HRT-FET cycles [1,2]. During the HRT cycle, a corpus luteum is not present, as the oestradiol treatment suppresses the natural cycle, and serum P4 only reflects the administered progesterone. The administration of additional progesterone from the time of blastocyst transfer in patients with low serum P4 in HRT-FET cycles has been shown to significantly enhance reproductive outcomes. This treatment regimen is referred to as the luteal phase support (LPS) rescue strategy, and it has been reported effective across various routes of progesterone administration, including oral, intramuscular (IM), subcutaneous (SC), and rectal formulations [3,4]. However, a fundamental aspect of the LPS rescue policy is that patients need to have a blood sample taken and analysed. This procedure, for some patients, can be a source of stress as well as discomfort, and it furthermore necessitates meticulous logistical organisation. Moreover, the validity of a single serum P4 level as an accurate representation of the patient’s progesterone status throughout the day remains unclear. It has previously been established that serum P4 levels exhibit circadian variation following vaginal administration, with considerable inter- and intra-individual variability [5].

Thus, it would be more convenient for patients if they could provide a urine sample which may represent the P4 levels throughout the day, potentially offering a more accurate assessment of P4 levels compared to a single serum P4 measurement.

Previous research suggests that serum concentrations of several steroid hormones and their urinary equivalents exhibit a high degree of correlation during the ovulatory cycle. Furthermore, several studies have found analysis of urine to be an effective method for detecting increases in pregnanediol 3-glucuronide (PDG) levels, which can confirm ovulation [6,7,8]. In addition, Roos et al. (2015) demonstrated a significant correlation between urine levels of PDG and serum P4 levels, suggesting that they can be used interchangeably [9].

PDG is a progesterone metabolite, and the analysis is often a cumbersome manual competitive enzyme-linked immunosorbent assay (ELISA), which is not accessible in a daily clinical fertility setting. Instead, Gifford et al. tested the possibility of analysing P4 levels in urine by validating two automated assay platforms [8]. The platforms validated were the automated P4 chemiluminescent microparticle immunoassay (Abbott Laboratories, Lake Bluff, IL, USA) and the automated P4 electrochemiluminescence immunoassay (Roche Diagnostics Ltd., Welwyn Garden City, UK). The results showed that the urinary P4 analysed by an automated assay platform was comparable to the PDG concentration in urine, measured by competitive PDG ELISA. Furthermore, the Abbott ARCHITECT system proved the most accurate urinary P4 analysis [8].

The primary objective of this study was to assess the possible correlation between the initial morning urine progesterone level and the serum progesterone level when serum sampling was performed within a narrow time frame between two and four hours following the administration of a standard vaginal progesterone dose in HRT-FET cycles. As a secondary objective we aimed to assess the use of urine P4 as a diagnostic tool to help determine patients in need of luteal phase rescue.

## 2. Results

There was a significant difference in median urine P4 levels between patients with serum P4 levels higher or lower than 11 ng/mL, 6400 ng/mL IQR [2528; 11,930] vs. 3408 ng/mL IQR [592; 6688], *p* < 0.001 (Table 1). Adjusting for creatinine (Cr) also showed a significant difference, 5108 ng/mg IQR [1831; 10,584] compared to 2459 ng/mg IQR [461; 6081], *p* < 0.001 (Table 1). Figure 1 displays a scatter plot illustrating the relationship between serum and urine progesterone levels. Each point represents an individual sample measuring both the serum and urinary concentrations. A line of best fit, derived from linear regression analysis, is included. In Figure 2 urine progesterone is adjusted for Cr. However, no correlation was found between urine P4 and serum P4 measured two to four hours after vaginal progesterone administration, Spearman’s rank correlation (rho) = 0.26, *p* < 0.0001 and Pearson’s correlation (r) = 0.10, *p* = 0.03. Adjusting urine P4 levels for Cr made no significant impact on the results, Spearman’s rank correlation coefficient (rho) = 0.26, *p* < 0.0001 and Pearson’s correlation (r) = 0.06, *p* = 0.22. A Spearman correlation close to zero suggests a negligible correlation between the two variables, indicating they are largely independent of each other in a monotonic sense. Other characteristics of the study cohort have previously been published [3].

### 2.1. Dilution Range

The sample dilution for P4 urine analysis encompassed a broad concentration spectrum, ranging from the undiluted ‘neat’ sample to a highly diluted ratio of 1:81,920. As detailed in Table 2, the distribution of samples across each dilution factor is presented.

### 2.2. Reproductive Outcome

A subgroup analysis was conducted on the cohort of patients having serum P4 levels higher than 11 ng/mL. This analysis consisted of the calculation of an optimal cut-off for urine P4 and for urine P4 adjusted for Cr to achieve a live birth, using the Youden factor. For urine P4, the highest sensitivity (70%) and specificity (42%) for live birth were urine P4 levels ≥ 4000 ng/mL; 48% (107/222) vs. 35% (45/130), *p* = 0.013 (Table 3). When adjusted for Cr, the optimal cut-off was ≥3141 ng/mg (sensitivity 70.4%, specificity 40.5%) with a live birth of 47% (107/227) vs. 36% (45/125) if urine P4:Cr was <3141 ng/mg, *p* = 0.044.

A comparison was made of LBR in patients with urine P4 levels more or less than 4000 ng/mL using logistic regression, with adjustments made for age at oocyte retrieval, BMI, blastocyst score, blastocyst age and serum oestradiol at blastocyst transfer day. The results showed that the odds ratio (OR) was 1.8 in the group with urine P4 ≥ 4000 ng/mL, 95% CI [1.067; 3.018], *p* = 0.028.

## 3. Discussion

In this cohort study, we found a significant difference in median urine P4 levels between patients with serum P4 higher or lower than 11 ng/mL; however, no correlation was seen between urine P4 and serum P4 measured two to four hours after vaginal progesterone administration. Furthermore, patients with a urine P4 level higher than 4000 ng/mL had a 1.8 times higher chance for a live birth compared to patients with a urine P4 below this cut-off.

### 3.1. Serum Progesterone vs. Urine Progesterone

No correlation between serum P4 and urine P4 was found in the present study. Interestingly, this result does not correspond to the findings of previous studies, generally reporting a more consistent relationship between serum and urinary P4 levels as well as serum P4 and urine PDG levels. However, it is important to note that the former studies were conducted primarily in cohorts of women during their natural menstrual cycle. The endogenous P4 in a natural menstrual cycle results in a pulsatile and dynamic pattern of secretion from the corpus luteum. Serum progesterone levels fluctuate throughout the day, and urinary progesterone and PDG levels provide an integrated reflection of this hormonal activity over time.

In contrast to the HRT-FET cycle in which only exogenous progesterone is in circulation [9,10]. This method of delivery leads to supraphysiological peak levels in serum shortly after administration, followed by a rapid decline due to metabolism. Our study protocol involved a single serum progesterone measurement taken 2–4 h after vaginal administration. This snapshot may not accurately reflect the overall progesterone exposure throughout the day.

Stanczyk et al. (1997) described a significant correlation between urine progesterone and urine PDG levels, and they also found urine P4 to be similar to serum P4 levels throughout the ovulatory cycle [10]. These findings were corroborated by Roos et al. (2015) [9], who reported that urine PDG and serum P4 exhibited a high degree of correlation. The authors proposed the potential for these parameters to be utilised interchangeably [9]. Stanczyk et al. (1997) furthermore included a small cohort of women, who were administered a short regimen of oral and IM progesterone [10]. In this cohort they found urine P4 to be as effective a clinical marker of progesterone treatment as urine PDG.

In the present study, the standard blood sampling procedure was conducted between two to four hours following the administration of vaginal progesterone. A single serum P4 level measurement following vaginal progesterone administration provides only a static measurement, potentially coinciding with the peak serum concentration. In contrast, urinary P4 assessment has the potential to provide a more integrated reflection of P4 exposure throughout the day, despite the influence of variation in relation to the administration time point. In this case, the analysis of first morning urine is of particular significance, as it represents the cumulative urinary progesterone excretion across several preceding hours and represents a cumulative excretion of progesterone. Therefore, it provides a more integrated assessment of progesterone exposure, mitigating the impact of short-term fluctuations associated with vaginal administration.

In the context of the present study, serum P4 concentrations—potentially measured at or near peak levels—are unlikely to directly correlate with the urinary excretion, which reflects a longer time interval following vaginal progesterone administration. This could explain the observed discrepancy between the findings reported by Stanczyk et al. (1997) and Roos et al. (2015), in which a close correlation between urine PGD/urine P4 and serum P4 was seen [9,10].

### 3.2. Urine Progesterone as a Diagnostic Tool

In patients exhibiting a serum P4 level ≥ 11 ng/mL, the optimal cut-off for live birth was calculated to be a urinary P4 level ≥ 4000 ng/mL, using the Youden factor. This value defines the optimal urine P4 threshold with the highest level of accuracy, as measured by the sensitivity and specificity to achieve a live birth in patients with a serum P4 level ≥ 11 ng/mL.

When a comparison was made between the LBR for patients with a urinary P4 level higher or lower than 4000 ng/mL, the odds ratio was 1.8 for patients with a urine P4 above the cut-off. Consequently, patients exhibiting urine P4 levels above 4000 ng/mL had an almost 1.8 times higher probability of live birth when compared to patients with urine P4 levels below the calculated cut-off. The rationale behind this phenomenon may be attributed to the observation that certain patients exhibit remarkably elevated serum P4 levels at the time of blood sampling, specifically within the two to four-hour period following vaginal progesterone administration. However, during the following hours, there is a marked decrease in serum levels, which may be attributable to a rapid metabolism resulting in diminished progesterone levels during the majority of the day.

These findings suggest that urine progesterone could be a valuable diagnostic tool to help identify cases where luteal phase rescue with additional exogenous progesterone is needed.

### 3.3. Need for Luteal Phase Rescue

Currently, luteal serum P4 levels determine the need for luteal phase rescue. The luteal phase rescue protocol used in the study was administered in patients with a serum P4 lower than 11 ng/mL on the day of blastocyst transfer [3]. Previous studies reported slightly lower cut-off levels to determine the need for rescue serum (P4 < 10 ng/mL) [4,11,12].

The higher cut-off utilised in the present study, a higher number of patients received rescue in comparison to other study cohorts. However, it could be hypothesised that even patients with serum P4 levels higher than 11 ng/mL may benefit of additional progesterone treatment. This hypothesis is based on the results of the present study, which showed that 130/352 of the patients with a serum P4 ≥ 11 ng/mL had a urine P4 < 4000 ng/mL and of these patients only 45 (35%) experienced a live birth (Table 3). In comparison a significantly higher LBR, 107/222 (48%) was observed in the patients with a urine P4 ≥ 4000 ng/mL (*p* = 0.013).

The findings of this study indicate that relying on serum P4 levels alone to determine the need for luteal phase rescue could result in some patients being sub-optimally treated. This opens the question of whether urine P4 should be regarded as a more effective diagnostic tool than serum P4, or whether both serum and urine should be considered when determining the treatment plan during HRT-FET cycles. However, utilising both serum P4 and urine P4 as determining factors would result in a rise in the overall cost of the treatment and could also be considered more inconvenient for patients.

### 3.4. Limitations

Due to the lack of previous research within the topic, finding comparable data was difficult. Several previous studies focused on the correlation between urine PDG and serum progesterone and how both values could be used to confirm ovulation. However, research on the use of urine progesterone values to evaluate luteal phase supplementation during HRT-FET is scarce.

In the present study five samples needed to be diluted more than 20,480 times to analyse urine P4 levels. High amounts of dilution could potentially affect the accuracy of the analysis and can furthermore lead to impaired reproducibility of the results. Furthermore, applying different dilution factors for each sample to obtain the absolute P4 concentration would be too cumbersome and is not suitable for clinical laboratory practice. However, in this study, the effective cut-off value was P4 ≥ 4000 ng/mL, which equates to a > 1:100 dilution. Further study is required, but the sample analysis protocol can be simplified by setting a single dilution factor based on the cut-off value.

### 3.5. Future Research

While further research is needed to fully understand the relationship between urine and serum P4 levels during exogenous hormone therapy in HRT-FET cycles, this study offers promising preliminary evidence for the use of urine progesterone measurements. Longitudinal data is required to determine if urine P4 is a reliable marker of progesterone exposure, and large-scale validation in diverse cohorts is essential to define optimal cut-off values for predicting reproductive outcomes. Future studies should also explore the utility of urine progesterone in other aspects of fertility treatment, such as monitoring compliance with progesterone administration or evaluating the impact of different progesterone formulations.

## 4. Materials and Methods

### 4.1. Study Design and Settings

This prospective cohort study was conducted at a public fertility clinic centre from January 2020 to November 2022, and urine analyses were performed using the stored urine samples in June 2023 at the BHF Centre for Cardiovascular Science, The Queen’s Medical Research Institute, The University of Edinburgh, UK.

The current study was a sub-study of an observational study by Alsbjerg et al. (2023) where optimising progesterone levels during HRT-FET using rectal progesterone administration was explored [3].

### 4.2. Participants

All patients underwent a standardised HRT-FET protocol as described in detail below. Inclusion criteria included an age range of 18 to 45 years, at least one autologous vitrified blastocyst for transfer, and a body mass index (BMI) ranging from 18.5 to 34 kg/m^2^. The exclusion criteria were endometrial thickness of less than 7 mm after 12–20 days of treatment with 6 mg of oestradiol, the presence of a leading follicle or corpus luteum, no blastocysts available for transfer after thawing, uterine abnormalities, oocyte donation and severe dysregulated chronic medical diseases. Importantly, all patients were aged between 20 and 40 years at the time of oocyte retrieval. On the day of blastocyst transfer, blood samples were collected from 464 patients two to four hours following vaginal progesterone administration, and prior to embryo transfer, each patient provided a urine sample from their first morning void [3]. The study flow chart is shown in Figure 3.

### 4.3. Treatment Protocols

The HRT-FET protocol involved administration of 6 mg of oestradiol valerate (Estrofem^®^, Novo Nordisk, Denmark) daily commencing on the second day of the cycle. In addition, vaginal micronised progesterone (400 mg Cyclogest^®^LD. Collins, UK) administration was initiated on the 12th to 20th day of treatment. Administration time was 7 a.m. and 7 p.m. In addition to urine samples provided by the patients, serum P4 levels were measured on the sixth day of vaginal progesterone treatment, two to four hours after administration. These serum P4 levels were used to determine the need for luteal phase rescue treatment. Patients with serum P4 levels of less than 11 ng/mL were treated with additional rectally administered progesterone starting on the day of transfer [3].

A pregnancy test using serum hCG was performed nine to eleven days after the transfer. In cases where a positive pregnancy test (hCG > 10 IU/L) was achieved, the rectal luteal phase rescue treatment continued until gestational week 8 + 0. The patients continued vaginal progesterone administration until gestational week 10 + 0 [3].

### 4.4. Sample Size Calculation

This study was a sub-study of an observational study exploring the role of optimising progesterone levels during HRT-FET. Given the exploratory nature of this study, a power calculation was not conducted.

### 4.5. Urine Progesterone Analysis

Urine samples were aliquoted in the Eppendorf tubes and stored at −80 degrees in the ultra-low temperature freezer until the analysis. Samples were defrosted in the fridge overnight. On the day of the analysis, samples were mixed by vortex for 10 s and left in the fridge for forty minutes before the analysis. Urine P4 and urine creatinine (Cr) were analysed simultaneously on an Abbott ARCHITECT ci4100 analyser (Abbott Laboratories, Abbott Park, IL, USA) using a proprietary serum assay kit, the ARCHITECT Progesterone (7K77-25) and the Creatinine2 (04S95-20) assay kit, respectively. The assay was calibrated within the concentration range of 0.0 ng/mL to 40 ng/mL, with an analytical sensitivity of ≤0.1 ng/mL. The samples were diluted 1 in 320 using ARCHITECT Progesterone Manual Diluent (07K77-50). Samples were further diluted when the results exceeded the calibrated range and analysed with the lower dilution ratio or neat samples when the diluted result read lower than 0.7 ng/mL (lowest calibration point). The sample dilution ranged from a neat sample to 1 in 81,920 (Table 2). The recovery of the target was within 80 to 120% across the dilution used when the results without the dilution factor were less than 40 and over 0.7 ng/mL. The detailed validation of urine sample analysis using the Abbott proprietary serum assay kit is published elsewhere [8]. The urine samples were analysed with one freeze–thaw cycle, and progesterone in urine samples is generally considered stable. The P4 value was standardised with the Cr value for data analysis and expressed as ng (P4)/mg (Cr).

### 4.6. Serum Progesterone Analysis

Serum progesterone (P4) levels were measured using a direct chemiluminescent assay (Atellica, Siemens, Munich, Germany), a standard procedure routinely employed in the local biochemistry laboratory. All analyses adhered strictly to the manufacturer’s guidelines. The assay’s dynamic range extended from 1.0 to 1908 nmol/L, with a within-laboratory coefficient of variation (CV) of approximately ±12% at a concentration of 6.3 nmol/L and about ±8% at 39 nmol/L. All blood samples were processed for progesterone determination promptly after collection.

### 4.7. Statistical Analysis

Categorical variables were compared using the chi-squared test or Fisher’s exact test when cell counts were low. Continuous variables were assessed for normality using quantile-quantile plots. Normally distributed data were compared using the *t*-test, while non-normally distributed data were compared using the Mann–Whitney U test. Equal variance was assessed using the F-test. The highest probability of achieving a live birth in relation to urine P4 levels in patients with a serum P4 level ≥ 11 ng/mL was determined using the Youden Index, defined as J = max (Sensitivity [c] + Specificity [c] − 1), where c is the cut-off point. A Youden Index value of 1 indicates a perfect test, whereas a value of 0 or below indicates that the test is not fit for purpose [13,14]. Logistic regression models, adjusting for potential confounding variables, were employed to evaluate the impact of urine P4 levels above or below the calculated cut-off on LBR, accounting for age at oocyte retrieval, BMI, blastocyst score and blastocyst age, as well as serum oestradiol level on the day of transfer. Pearson’s correlation coefficient (r) and Spearman’s rank correlation coefficient (rho) were employed to assess the correlation between urine P4 and serum P4 levels. A *p*-value of <0.05 was considered statistically significant. Statistical analyses were performed using SPSS version 20 (IBM Corporation, New York, NY, USA) and STATA version 13 (StataCorp LLC, College Station, TX, USA).

### 4.8. Ethics

The study was approved by the Regional Ethical Committee, the Danish Data Protection and the Danish Medicines Agency on the 10 October 2019 (EudraCT no.: 2019-001539-29). The Good Clinical Practice (GCP) unit, Aarhus University, monitored the study, GCP protocol number: 787/2019.

## 5. Conclusions

A significant difference in median urine P4 levels between patients with serum P4 higher or lower than 11 ng/mL was seen; however, no correlation was found between urine P4 and serum P4 measured two to four hours after vaginal progesterone administration. Patients with a urinary P4 higher than 4000 ng/mL had a 1.8 times higher chance of a live birth compared to patients with a urine P4 below this cut-off. These findings suggest that urine progesterone could be a valuable diagnostic tool to help evaluate the need for luteal phase rescue treatment.

This study represents a novel and important step in exploring urine progesterone measurement in HRT-FET cycles and its correlation with reproductive outcomes. We believe that our findings set the groundwork for future, larger-scale studies that can further validate these initial observations.

However, further research is required in order to compare the diagnostic value of urine progesterone and serum P4 in HRT-FET cycles.

## Figures and Tables

**Figure 1 ijms-26-10795-f001:**
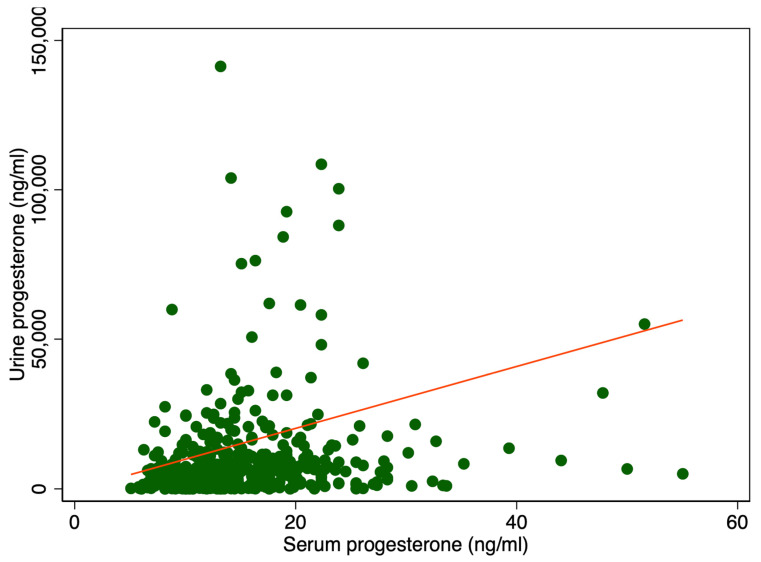
Serum P4 levels in relation to urine P4 levels. The red line represents the regression line or line of best fit.

**Figure 2 ijms-26-10795-f002:**
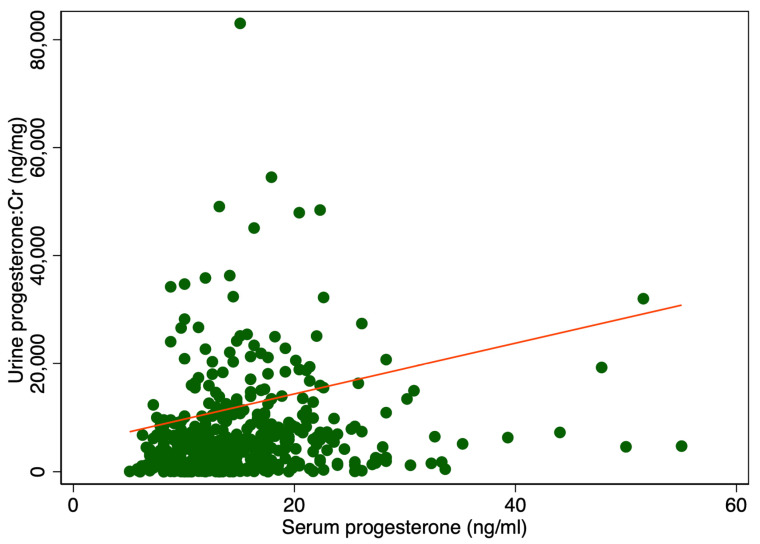
Serum levels in relation to urine progesterone levels adjusted for creatinine. The red line represents the regression line or line of best fit.

**Figure 3 ijms-26-10795-f003:**
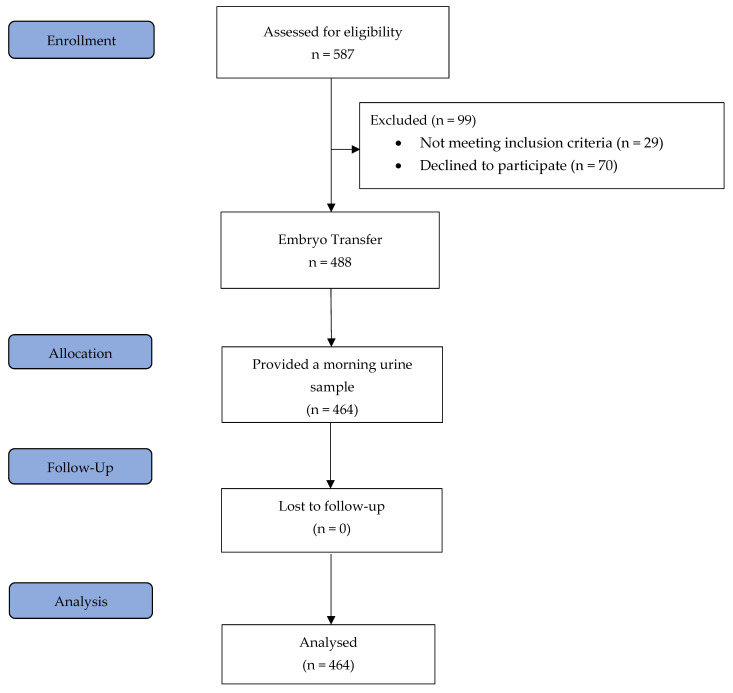
Study flow diagram.

**Table 1 ijms-26-10795-t001:** Patients and urine sample characteristics in relation to serum progesterone levels more or less than 11 ng/mL. IQR: interquartile range. SD: standard deviation. P4: Progesterone. Cr: Creatinine. P4:Cr: Progesterone adjusted for creatinine. ^1^ *t*-test. ^2^ Median test (independent samples median *t*-test and Mann–Whitney U test).

	All	Serum P4 Levels ≥ 11 ng/mL	Serum P4 Levels < 11 ng/mL	*p*-Value
**Number of samples**	464	352	112	
**Age, year**	31.5 ± 4.4	31.5 ± 4.4	31.7 ± 4.7	0.69 ^1^
**BMI, kg/m^2^**	25.1 ± 3.5	24.9 ± 3.6	26.1 ± 3.4	0.002 ^1^
**Median times for urine sample dilution, (range) [IQR]**	320 (1–81,920) [320; 2540]	320 (10–81,920) [320; 2560]	320 (1–2560) [320; 560]	<0.025 ^2^
**Median urine P4, ng/mL, (range) [IQR]**	5632 (0.8–1,081,344) [1888; 11,176]	6400 (10–1,081,344) [2528; 11,930]	3408 (1–59,904) [592; 6688]	<0.001 ^2^
**Mean urine Cr, mg/mL ± SD**	1.36 ± 0.57	1.37 ± 0.58	1.31 ± 0.52	0.37 ^1^
**Median urine P4:Cr, ng/mg, (range) [IQR]**	4562.47 (1–994,046) [1496; 9086]	5108 (5–994,046) [1831; 10,584]	2459 (1–34,710) [461; 6081]	<0.001 ^2^

**Table 2 ijms-26-10795-t002:** Distribution of samples across the different dilutions.

Dilution	0	10	160	320	640	1280	2560	5120	20,480	40,960	81,920	Total
**Number of samples**	2	35	20	249	31	2	111	10	2	1	3	**466**

**Table 3 ijms-26-10795-t003:** Comparison of live birth rate (LBR) according to serum progesterone and urine progesterone. P4: Progesterone. * Patients with serum P4 < 11 ng/mL were treated with luteal phase rescue (additional rectal progesterone administration after blood sampling and urine sampling). ** *p*-values were calculated to compare results between urine P4 levels < 4000 ng/mL and ≥4000 ng/mL.

			Urine P4 10–100 ng/mL	Urine P4 100–1000 ng/mL	Urine P4 1000–2000 ng/mL	Urine P4 2000–3000 ng/mL	Urine P4 3000–4000 ng/mL	Urine P4 10–4000 ng/mL	Urine P4 ≥4000 ng/mL	*p*-Value **
**Serum P4** **≥11 ng/mL**	**Live birth** **n (%)**	No	9	25	14	19	18	85 (65)	115 (52)	0.013
		Yes	4	13	13	8	7	45 (35)	107 (48)	
	*Total*		*13*	*38*	*27*	*27*	*25*	*130*	*222*	
**Serum P4** **<11 ng/mL** *	**Live birth** **n (%)**	No	5	15	5	6	5	36 (58)	25 (50)	0.394
		Yes	3	11	5	4	3	26 (42)	25 (50)	
	*Total*		*8*	*26*	*10*	*10*	*8*	*62*	*50*	

## Data Availability

The raw data supporting the conclusions of this article will be made available by the authors on request.

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
