# Peer review of "Urine Progesterone Level as a Diagnostics Tool to Evaluate the Need for Luteal Phase Rescue in Hormone Replacement Therapy Frozen Embryo Transfer Cycles"

_ijms, 2025, doi:10.3390/ijms262110795_

Round 1
Reviewer 1 Report
Comments and Suggestions for Authors
In present work, Hansen et al. try to study the use of urine progesterone as a diagnostic tool during the luteal phase. This study showed that that urine progesterone could be a valuable diagnostic tool to evaluate the need for additional progesterone in hormone replacement therapy frozen embryo transfer cycles. However, there are some questions that should be explained, and the data volume of this study is not enough for published in IJMS.
Major concerns
- This study was a sub-study of an observational study using rectal progesterone dministration was explored. Only two figures and three simple tables include serum levels and urine progesterone levels adjusted for creatinine or not. In addition, reference numbers are only 14. Furthermore, in this study, vaginal progesterone (a conventional method) administration is used.
- The size of corepus luteum may be close related to the serum levels in relation to urine progesterone levels. The data for the size of corepus luteum detect by an ultrasound scan are needed.
- The detail for Materials and methods are needed. How to perform pregnancy testing and vaginal progesterone? Why blood samples were only collected two to four hours following vaginal progesterone administration and prior to embryo transfer? In addition, two to four hours is very long. The half-life of progesterone (the time required for half metabolism) is approximately 5-20 hours. Therefore, two or four hours has significant effect on the serum levels and urine progesterone.
- Line 228, ‘18 and 45 years of age’, the age difference is too large. In general, women over the age of 35 will experience a rapid decline in their fertility.
- English grammar and writing should be checked and revised throughout the manuscript.
Minor concerns
- title, delete ‘(HRT-FET)’ and ‘ - a prospective observational study’.
- Lines 21 and 22, correct ‘6mg’ and ‘400mg’ to ‘6 mg’ and ‘400 mg’. Please check it throughout the manuscript.
- Line 26, ‘p<0.001’. ‘p’ should be italic. Please check it throughout the manuscript.
- ‘Keywords’ should be corrected. Some are not Keywords.
- Lines 81 and 82, ‘p<0.001’ or ‘P<0.001’ ?
- Lines 128-141, different fonts.
- Lines 156-173, there is no a reference.
- Lines 310-322, some words with large size.
The English could be improved to more clearly express the research.
Author Response
Answer to reviewer
We thank You for the comments. The questions and answers are listed below.
Reviewer 1#
In present work, Hansen et al. try to study the use of urine progesterone as a diagnostic tool during the luteal phase. This study showed that that urine progesterone could be a valuable diagnostic tool to evaluate the need for additional progesterone in hormone replacement therapy frozen embryo transfer cycles. However, there are some questions that should be explained, and the data volume of this study is not enough for published in IJMS.
Major concerns
This study was a sub-study of an observational study using rectal progesterone administration was explored. Only two figures and three simple tables include serum levels and urine progesterone levels adjusted for creatinine or not. In addition, reference numbers are only 14. Furthermore, in this study, vaginal progesterone (a conventional method) administration is used.
Answer:
We are grateful to the reviewer for his insightful observation regarding the data quantity in our study. It is indeed the case that the sample size appears limited; however, this is indicative of the pioneering nature of the research. It is important to note that this is the first publication to investigate urine progesterone levels in hormone replacement therapy (HRT) frozen embryo transfer (FET) cycles and to examine their correlation with reproductive outcomes such as serum progesterone levels and live birth rates.
As such, the current study provides initial insights and establishes a foundation for future research in this area. The limited references cited are primarily due to the novelty of this investigation—there are few, if any, previous studies that have explored urine progesterone measurements in this specific clinical context, making our publication a pioneering contribution to the field.
Further, the characteristics of the cohort, including demographic and clinical parameters, have been detailed in a prior publication, which we have referenced accordingly, to avoid redundancy and to contextualise the current findings within the broader scope of our research.
While additional data could enhance statistical power and generalisability, the present findings are valuable as preliminary evidence which underscores the potential utility of urine progesterone as a non-invasive marker in HRT-FET cycles. We hope that future studies including larger cohorts will build upon these initial observations to further validate and expand our results.
The size of corepus luteum may be close related to the serum levels in relation to urine progesterone levels. The data for the size of corepus luteum detect by an ultrasound scan are needed.
Answer:
The present study employs a hormone replacement therapy (HRT) protocol. Consequently, a corpus luteum is not present in women undergoing estradiol treatment, as their natural cycle is suppressed. One of the exclusion criteria was if a patient developed a leading follicle or a corpus luteum; inclusion and exclusion criteria have now been added. Line 259-265.
The detail for Materials and methods are needed. How to perform pregnancy testing and vaginal progesterone?
Answer:
The pregnancy test was a serum test, and the cutoff value for a positive test was >10IU/L. This has been incorporated. Furthermore, the timing of vaginal progesterone administration has been added. Line 279-289.
Why blood samples were only collected two to four hours following vaginal progesterone administration and prior to embryo transfer? In addition, two to four hours is very long. The half-life of progesterone (the time required for half metabolism) is approximately 5-20 hours. Therefore, two or four hours has significant effect on the serum levels and urine progesterone.
Answer:
We acknowledge that more blood sampling would have been preferable; however, it would have been difficult to finalise a study including 464 patients in a clinical setting. We decided to measure serum progesterone (P4) levels between two and four hours after vaginal progesterone administration, as this is the period during which serum P4 levels are highest. This level is used to determine whether patients require additional progesterone administration. Furthermore, urine was only collected once during treatment, on the day of blastocyst transfer, when serum P4 was measured.
- '18 and 45 years of age', the age difference is too large. In general, women over the age of 35 will experience a rapid decline in their fertility.
Answer:
The patients are recruited from a public fertility clinic, and we treat patients aged 18–45. However, the age range for oocyte retrieval was between 20 and 40 years. This has been added. Line 265.
English grammar and writing should be checked and revised throughout the manuscript.
Answer:
The manuscript has been proofread for spelling and grammatical errors.
Minor concerns
title, delete '(HRT-FET)' and ' - a prospective observational study'.
Answer: the title has been changed.
Lines 21 and 22, correct '6mg' and '400mg' to '6 mg' and '400 mg'. Please check it throughout the manuscript.
'Keywords' should be corrected. Some are not Keywords.
Lines 81 and 82, 'p<0.001' or 'P<0.001' ?
Lines 128-141, different fonts.
Lines 310-322, some words with large size.
Lines 156-173, there is no a reference.
Answer:
We thank the reviewer for pointing out all minor concerns.
The keywords have been reduced.
The missing reference is [10] Stanczyk, F.Z., et al., Urinary progesterone and pregnanediol. Use for monitoring progesterone treatment. J Reprod Med, 1997. 42(4): p. 216-22.
All other issues have been addressed.
In conclusion, this study represents a novel and important step in exploring urine progesterone measurement in HRT-FET cycles and its correlation with reproductive outcomes. We believe that our findings set the groundwork for future, larger-scale studies that can further validate these initial observations. We have thoroughly addressed all issues and questions raised by the reviewers and hope that the manuscript is suitable for publication, contributing meaningfully to this developing field of research.
We would like to express our gratitude to the reviewer for valuable comments. The questions and comments have been answered and we hope that the manuscript, after these changes, is suitable for publication.
Reviewer 2 Report
Comments and Suggestions for Authors
This is a well-designed addressing interesting clinical questions.
However, the manuscript could benefit from clarification and refinement before it is suitable for publication.
- The current order of sections is unconventional. Specifically, the Results appear before Material and Methods, perhaps by mistake. Reorder the manuscript into the standard flow (Introduction – Methods – Results – Discussion – Conclusion).
- The description of the laboratory methods for the urine and serum progesterone analysis is excessively detailed. Make this part of the text more concise and provide the extra information in a table or a supplementary file.
- A central premise of this study is the potential use of urinary progesterone levels as a diagnostic alternative to serum measurements for guiding luteal phase support. However, the lack of correlation between urine and serum progesterone, a key finding in your results, raises critical questions about this substitution. At present, the explanation provided in the Discussion is unsatisfactory. It is speculative and somewhat arbitrary. A clearer account of the differences between the endogenous luteal function and the exogenous hormone delivery is perhaps needed.
A few stylistic issues in the text (e.g., avoid small paragraphs. repeated phrases, overuse of passive voice) could be addressed during revision.
Author Response
Answer to reviewer
We thank You for the comments. The questions and answers are listed below.
Reviewer 2#
This is a well-designed addressing interesting clinical questions.
However, the manuscript could benefit from clarification and refinement before it is suitable for publication.
The current order of sections is unconventional. Specifically, the Results appear before Material and Methods, perhaps by mistake. Reorder the manuscript into the standard flow (Introduction – Methods – Results – Discussion – Conclusion).
Answer:
We agree with Reviewer 2 that this order is not widely used. However, it is the order described in the 'Instructions for Authors' section of the IJMS website.
The description of the laboratory methods for the urine and serum progesterone analysis is excessively detailed. Make this part of the text more concise and provide the extra information in a table or a supplementary file.
Answer:
We agree with Reviewer 2. However, before this manuscript was accepted for peer review, we were asked to provide more information on this issue by the journal as the journal focusses on molecular science.
A central premise of this study is the potential use of urinary progesterone levels as a diagnostic alternative to serum measurements for guiding luteal phase support. However, the lack of correlation between urine and serum progesterone, a key finding in your results, raises critical questions about this substitution. At present, the explanation provided in the Discussion is unsatisfactory. It is speculative and somewhat arbitrary. A clearer account of the differences between the endogenous luteal function and the exogenous hormone delivery is perhaps needed.
Answer:
We have provided a clearer account for the differences between endogenous and exogenous progesterone dynamics in the discussion. Lines 156-182. Furthermore, we acknowledge that further research is needed to fully elucidate the relationship between urine and serum progesterone in HRT-FET cycles. Future studies should consider measuring serum progesterone levels at multiple time points throughout the day to capture the full profile of progesterone exposure and larger-scale clinical trials are needed to validate the utility of urine progesterone as a diagnostic tool for guiding luteal phase support. Nonetheless, this pioneering study suggests the use of urine progesterone as a diagnostic parameter in HRT-FET and can serve as a guide for future investigations.
Comments on the Quality of English Language
A few stylistic issues in the text (e.g., avoid small paragraphs. repeated phrases, overuse of passive voice) could be addressed during revision.
Answer:
The manuscript has been proofread, focusing on the issues mentioned.
In conclusion, this study represents a novel and important step in exploring urine progesterone measurement in HRT-FET cycles and its correlation with reproductive outcomes. We believe that our findings set the groundwork for future, larger-scale studies that can further validate these initial observations. We have thoroughly addressed all issues and questions raised by the reviewers and hope that the manuscript is suitable for publication, contributing meaningfully to this developing field of research.
We would like to express our gratitude to the reviewer for valuable comments. The questions and comments have been answered and we hope that the manuscript, after these changes, is suitable for publication.
Reviewer 3 Report
Comments and Suggestions for Authors
The manuscript entitled "Urine progesterone level as a diagnostics tool to evaluate the need for luteal phase rescue in Hormone Replacement Therapy Frozen Embryo Transfer (HRT-FET) cycles - a prospective observational study" is written in clear understandable English and focuses the attention of the reader on the problems of progesterone measuring (urine and blood), differences of these two approaches of measuring and also authors discuss the outcomes (livebirths) across the patinets with different progesterone levels.
I have few commentaries for the authors.
Major
- You used P4 obtained from urine in your analysis. What is stability of P4 in urine samples? Did you performed calibration lines? Is it stable during freezing and transportation?
-
Statistics section: The statistical tests employed were the chi-squared test, the t-test and the Mann- Whitney-U test. Explain where and why you used each type of tests.
-
«This hypothesis is based on the results of the present study, 183 which showed that 130/352 of the patients with a serum P4 ≥11 ng/ml had a urine P4» So significant differences in P4 levels across the group. I am sure that livebirths were across the patients with P4 3000-4000. Provide more detailed information about this group. Make a subgroups: 11-100, 100-1000, 1000-2000, 2000-3000, 3000-4000.
-
Lines 229-232: “On the day of blastocyst transfer, blood samples were collected from 464 patients two to four hours following vaginal progesterone administration, and prior to embryo transfer each patient provided a urine sample from their first morning void”. Patients collected their urine before or after vaginal progesterone consumption?
- The manuscript contains only 14 refereces. I think that authors may enhance Discussion and Introduction sections.
Minor
1. Line 124 “a 1.8 times higher change” maybe chance?
Author Response
Answer to reviewer
We thank You for the comments. The questions and answers are listed below.
Reviewer 3#
The manuscript entitled "Urine progesterone level as a diagnostics tool to evaluate the need for luteal phase rescue in Hormone Replacement Therapy Frozen Embryo Transfer (HRT-FET) cycles - a prospective observational study" is written in clear understandable English and focuses the attention of the reader on the problems of progesterone measuring (urine and blood), differences of these two approaches of measuring and also authors discuss the outcomes (livebirths) across the patinets with different progesterone levels.
I have few commentaries for the authors.
Major
You used P4 obtained from urine in your analysis. What is stability of P4 in urine samples? Did you performed calibration lines?
Answer:
Answer:
The assay was calibrated with the two calibrators (0.7 and 40.0 ng/mL). The ARCHITECT Progesterone assay is designed to have a slope of 0.8 to 1.2, inclusive, and a correlation coefficient of ≥0.95 when compared to a commercially available assay
Is it stable during freezing and transportation?
The progesterone in urine samples is generally considered stable, and we thawed at 4 degrees overnight, and this method was used previously (ref [8]). Gifford et al. tested the stability of urine progesterone on the Abbott ARCHITECT analyser in the same facility. They did not find a significant reduction after the freeze-thaw cycles (ref [8]). We analysed samples with one freeze-thaw cycle, and the samples were shipped safely without thawing. Line 303-305.
Statistics section: The statistical tests employed were the chi-squared test, the t-test and the Mann- Whitney-U test. Explain where and why you used each type of tests.
Answer:
We used the chi-squared test to compare categorical outcomes such as live birth rates. We used the t-test to compare the means of normally distributed continuous variables and the Mann–Whitney U test for non-normally distributed continuous variables. The choice depended on the type of data and the distribution assumptions. The statistics paragraph has been updated accordingly. See lines 312–326.
«This hypothesis is based on the results of the present study, 183 which showed that 130/352 of the patients with a serum P4 ≥11 ng/ml had a urine P4» So significant differences in P4 levels across the group. I am sure that livebirths were across the patients with P4 3000-4000. Provide more detailed information about this group. Make a subgroups: 11-100, 100-1000, 1000-2000, 2000-3000, 3000-4000.
Answer:
As requested, Table 3 has been expanded to include subgroups of varying urine progesterone levels. A trend for a low live birth rate in all urine progesterone subgroups was identified, despite the fact that patients had a serum progesterone level higher than 11 ng/ml.
Lines 229-232: “On the day of blastocyst transfer, blood samples were collected from 464 patients two to four hours following vaginal progesterone administration, and prior to embryo transfer each patient provided a urine sample from their first morning void”. Patients collected their urine before or after vaginal progesterone consumption?
Answer:
All patients were instructed to administer vaginal progesterone at 7am and to collect the first morning urine voiding sample as mentioned in the M and M section. Lines 268 and 274. The present study has no data on which patients were administered progesterone either prior to or following the collection of urine.
The manuscript contains only 14 refereces. I think that authors may enhance Discussion and Introduction sections.
Answer:
The limited references cited are primarily due to the novelty of this investigation—there are few, if any, previous studies that have explored urine progesterone measurements in this specific clinical context, making our publication a pioneering contribution to the field.
The discussion has been enhanced by the provision of a clearer account of the differences between endogenous and exogenous progesterone dynamics. Furthermore, the potential benefits of using urine progesterone despite the lack of direct correlation have been elaborated. Finally, recommendations for future research have been provided in order to address the remaining uncertainties.
Minor
Line 124 “a 1.8 times higher change” maybe chance?
Answer:
The spelling mistake has been corrected.
In conclusion, this study represents a novel and important step in exploring urine progesterone measurement in HRT-FET cycles and its correlation with reproductive outcomes. We believe that our findings set the groundwork for future, larger-scale studies that can further validate these initial observations. We have thoroughly addressed all issues and questions raised by the reviewers and hope that the manuscript is suitable for publication, contributing meaningfully to this developing field of research.
We would like to express our gratitude to the reviewer for valuable comments. The questions and comments have been answered and we hope that the manuscript, after these changes, is suitable for publication.
Round 2
Reviewer 1 Report
Comments and Suggestions for Authors
Thanks for author’s responses. However, there are some questions that should be explained, and the data volume of this study is not enough for published in IJMS.
- This study was a sub-study of an observational study using rectal progesterone dministration was explored. Only two figures and three simple tables include serum levels and urine progesterone levels adjusted for creatinine or not. In addition, reference numbers are only 14. The data volume of this study is not enough for published in IJMS. I do not agree with the authors’ response.
- The first look of this revised manuscript impressed me is a mess. There are lot of different types and size of words and sentences, which does not write based on this journal style.
- Most of all, in page 10, authors state that ‘A pregnancy test using serum hCG was performed nine to eleven days after the transfer’. However, pregnancy test has been performed using urine hCG, which is very common at present.
- This reviewer suggested that the size of corepus luteum may be close related to the serum levels in relation to urine progesterone levels. The data for the size of corepus luteum detect by an ultrasound scan are needed.
However, authors respond that the present study employs a hormone replacement therapy (HRT) protocol. Consequently, a corpus luteum is not present in women undergoing estradiol treatment, as their natural cycle is suppressed.
Nevertheless, no a corpus luteum will lead to almost no P4 production. How to define the luteal phase? Why suppress P4 production by corpus luteum using estradiol treatment during embryo transfer.
- There are lot of low wrongs. For example,
Page 3, ‘they be used interchangeably [9]’.
Page 3, pregnanediol 3-glucuronide (PDG); Page 6, pregnanediol glucuronide (PDG).
Page 9, ‘Alsbjerg, B. et al. (2023)’ should change to ‘Alsbjerg et al. (2023)’.
Comments on the Quality of English LanguageThe English could be improved to more clearly express the research.
Author Response
Thanks for author’s responses. However, there are some questions that should be explained, and the data volume of this study is not enough for published in IJMS.
This study was a sub-study of an observational study using rectal progesterone dministration was explored. Only two figures and three simple tables include serum levels and urine progesterone levels adjusted for creatinine or not. In addition, reference numbers are only 14. The data volume of this study is not enough for published in IJMS. I do not agree with the authors’ response.
Answer:
We acknowledge that this study only includes 464 urine samples, and that only progesterone and creatinine analyses have been performed. However, we cannot increase the amount of data at this point, as the study has been finalised. Notably, this is the first study to evaluate urine progesterone in relation to HRT-FET cycles.
The first look of this revised manuscript impressed me is a mess. There are lot of different types and size of words and sentences, which does not write based on this journal style.
Answer:
We apologise for our inability to comprehend the aforementioned comment. It is evident that the style employed in the uploaded and downloaded PDF files from the website is consistent. Heading styles are presented in bold font, while table and figure texts are rendered in italics. The body text is set in regular font, employing the Calibri 12 font.
Most of all, in page 10, authors state that ‘A pregnancy test using serum hCG was performed nine to eleven days after the transfer’. However, pregnancy test has been performed using urine hCG, which is very common at present.
Answer:
We apologise for our inability to comprehend the query posed. As stated in the material and methods section, the pregnancy test was performed on a serum sample, rather than a urine sample.
This reviewer suggested that the size of corepus luteum may be close related to the serum levels in relation to urine progesterone levels. The data for the size of corepus luteum detect by an ultrasound scan are needed.
However, authors respond that the present study employs a hormone replacement therapy (HRT) protocol. Consequently, a corpus luteum is not present in women undergoing estradiol treatment, as their natural cycle is suppressed. Nevertheless, no a corpus luteum will lead to almost no P4 production. How to define the luteal phase? Why suppress P4 production by corpus luteum using estradiol treatment during embryo transfer.
Answer:
We disagree and need to repeat our answer. It is basic physiology that, in Hormone Replacement Therapy, no corpus luteum is present. Meaning that it does not make any sense to measure the size by ultrasound scan, as it is non-existent. The HRT-FET cycle is mimicking the natural cycle, with first estradiol treatment and thereafter vaginal progesterone is added, which mimics the luteal phase. The treatment protocol has been described in lines 271-274.
There are lot of low wrongs. For example,
Page 3, ‘they be used interchangeably [9]’.
Page 3, pregnanediol 3-glucuronide (PDG); Page 6, pregnanediol glucuronide (PDG).
Page 9, ‘Alsbjerg, B. et al. (2023)’ should change to ‘Alsbjerg et al. (2023)’.
Answer:
The typos have been corrected. With regard to the English language, the text has been thoroughly proofread. Furthermore, one of the co-authors is a native speaker from Scotland and is able to write and speak English fluently. I can confirm that the manuscript has been accepted by another reviewer and that they are satisfied with the standard of English. We do not wish to make any further corrections.
Reviewer 3 Report
Comments and Suggestions for Authors
Can be accepted.
Author Response
Thank you for accepting our answer and the submitted manuscript.